# LSTM input timestep optimization using simulated annealing for wind power predictions

**Muhammad Muneeb** *

Department of Electrical Engineering and Computer Science, Khalifa University of Science and Technology, Abu Dhabi, United Arab Emirates

* muneebsiddique007@gmail.com

**Data Availability Statement:** Data are available at https://github.com/MuhammadMuneeb007/LSTM-window-size-optimization-using-simulated-annealing-for-wind-power-prediction/tree/main/NaiveLSTM.

## Abstract

Wind energy is one of the renewable energy sources like solar energy, and accurate wind power prediction can help countries deploy wind farms at particular locations yielding more electricity. For any prediction problem, determining the optimal time step (lookback) information is of primary importance, and using information from previous timesteps can improve the prediction scores. This article uses simulated annealing to find an optimal time step for wind power prediction. Finding an optimal timestep is computationally expensive and may require brute-forcing to evaluate the deep learning model at each time. This article uses simulated annealing to find an optimal time step for wind power prediction. The computation time was reduced from 166 hours to 3 hours to find an optimal time step for wind power prediction with a simulated annealing-based approach. We tested the proposed approach on three different wind farms with a training set of 50%, a validation set of 25%, and a test set of 25%, yielding MSE of 0.0059, 0.0074, and 0.010 for each wind farm. The article presents the results in detail, not just the mean square root error.

## Introduction

Countries are moving towards renewable energy sources [1–3] due to the recent increase in global warming, and sources like solar energy and wind energy can play a crucial role in reducing carbon dioxide emissions in the environment. Wind power is one energy source that can help generate free electricity without pollutants. Wind power depends on the location where windmills are installed and on wind speed and direction, and in these articles [4–7], researchers illustrated the features which can be used for wind power prediction.

This article focuses on improving the performance of a deep learning algorithm for wind power prediction and reducing computational time. Previous timesteps are used to predict the current timestep for any prediction problem [8, 9]. For instance, information from timestep 1,2 and 3 can be combined with timestep 4 to predict the wind power for timestep 4, but finding the optimal lookback is difficult because we do not know how much past information should be included to make the current prediction. There are many ways to find that

**Funding:** This research work is not funded by any organization, but Khalifa University will cover the open access publication fee. Second, the funders had no role in study design, data collection and analysis, decision to publish, or preparation of the manuscript. Third, the author received no specific funding for this work.

**Competing interests:** The authors declare that no competing interests exist.

information. For instance, we can use the brute-force technique from lookback = 1 to 500 and run LSTM for each lookback. There are several issues associated with this technique. Firstly, as we increase the lookback variable, the data size grows exponentially, resulting in an exponential increase in training time. Secondly, we do not know how long we should train the LSTM model to obtain optimal results [10–12]. Thirdly, the optimal time step for the training data may not be optimal for the test set. These issues raise concerns for a better approach in which we can find an optimal lookback or how many previous timestep information should be used to predict the current timestep, and for how long we should train the machine learning model to yield an optimal performance (formally known as a number of epochs).

In this paper, we used a simulated annealing-based [13] + LSTM approach for wind power prediction, allowing us to find an optimal lookback in a limited number of epochs resulting in reduced training time.

The paper focuses on wind power prediction, but it is important to notice that it can also be used for other predictions. The previous timestep inclusion to predict the current timestep can significantly improve the performance. Optimization algorithms like particle swarm optimization [14], genetic algorithm, and hill-climbing can be used to find an optimal lookback. We used simulated annealing, which improves the current results generated using a genetic algorithm.

The important point to notice here is that the contribution of this article is not just using the simulated annealing with LSTM for wind power prediction but also how we used it to find an optimal look back, and it involves running some other computational steps which are documented in the methodology.

In these papers [15–17], researchers compared wind power prediction based on physical, statistical, and hybrid methods over different time scales.

There are many algorithms for wind power prediction like auto-regression moving average model [18], LSTM with particle swarm optimization [19], LSTM with extrapolation capability [20], double-stage hierarchical approach for energy management [21], and Cluster-Based Ensemble Regression models [22].

Following are the scientific contribution of this research work.

- We integrated simulated annealing (optimization algorithm) and LSTM (time series forecasting algorithm) to find an optimal lookback to reduce the computation time, find an optimal lookback, and improve the forecasting performance.

- The proposed integration is also valid for other deep learning algorithms like BILSTM, GRU, and machine learning-based regression algorithms with minor modifications. It is also one of the future directions that we considered for optimization.

## Materials and methods

This section explains the preprocessing performed on the dataset, the difference between three LSTM models, and the integration of simulated annealing with LSTM for wind power prediction.

### Dataset division

The dataset we used for the analysis is available on this link. There are about 16 features and one power variable to predict.

The training set is 50%, the test set is 25%, and the validation is 25%. The next step is to transform the dataset based on the lookback parameter, which shows how many previous

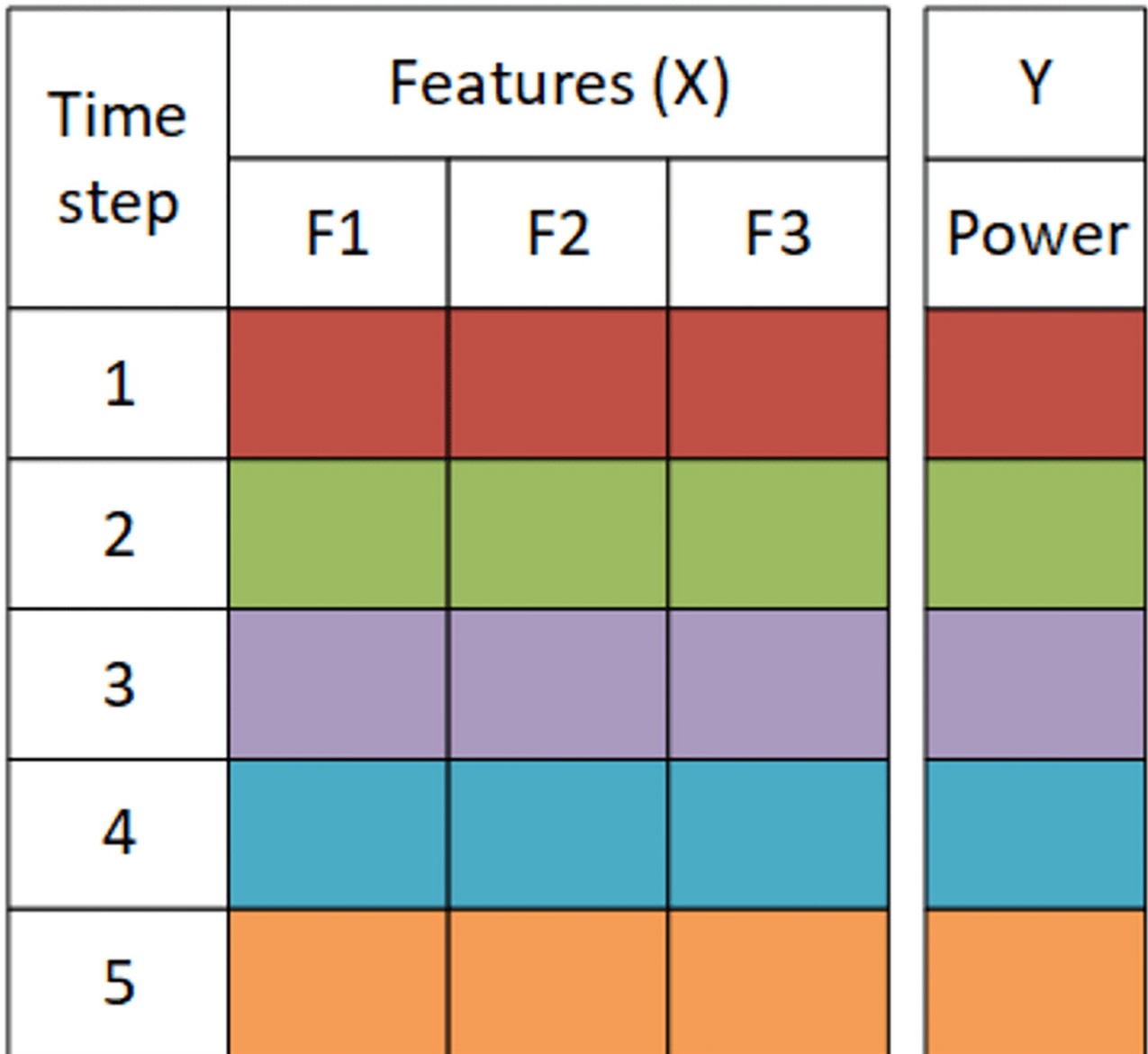

**Fig 1. This figure illustrates the form of the dataset for lookback = 1.**

time step information is to be included to make the current prediction. The transformation plays a key role in the whole process, and it can drastically increase the size of the dataset and computation time. To understand the dataset transformation step refer to Figs 1–3. In general, we do not know how many previous time step information should be included for the current time prediction, so an efficient approach like simulated annealing is required to find that optimal lookback. Consider Fig 1 each row represents the feature values at each time step. In Fig 2 the data is transformed to 4 time step. In Fig 3 data is transfored to 3 time step.

The transformation step for any number of lookbacks is explained in the code. What follows describes the simulated annealing and LSTM working.

| Time step | Features (X) | | | | | | Y |
|---|---|---|---|---|---|---|---|
| | F1 t-1 | F3 t-1 | F3 t-1 | F1 t | F2 t | F3 t | Power |
| 1 | | | | | | | |
| 2 | | | | | | | |
| 3 | | | | | | | |
| 4 | | | | | | | |

**Fig 2. This figure illustrates the form of the dataset for lookback = 2.**

## Simulated annealing

Simulated Annealing (SA) is an optimization technique for locating global optima. Simulated annealing employs the objective function of an optimization problem, which in our case is the MSE. The root mean square or explained variance can also be used as an optimization function, but we used the mean square error.

The method works similarly to a hill-climbing algorithm; instead of only choosing the optimal step, it chooses a random move. It constantly changes the current solution if the chosen move improves the solution. Otherwise, the procedure will proceed with a probability of fewer than one. With the "badness" of the maneuver, the chance of changing the current solution falls exponentially.

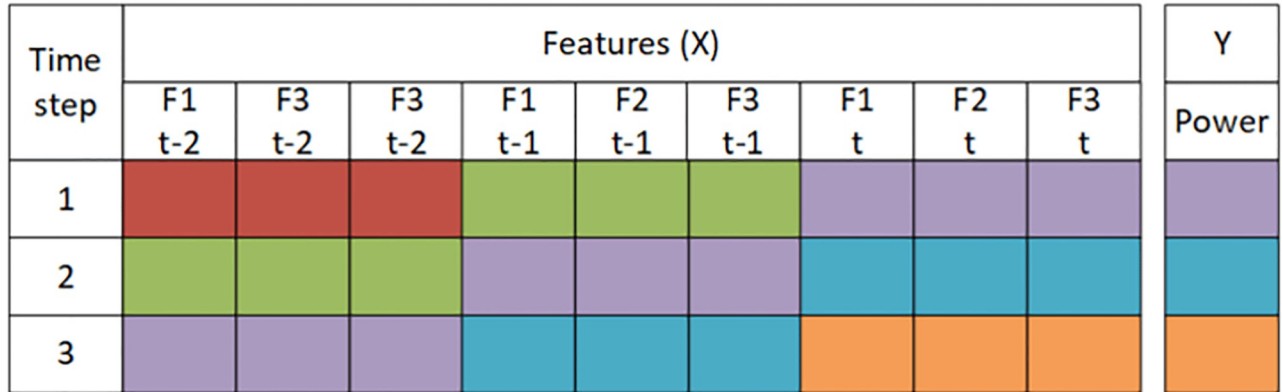

**Fig 3. This figure illustrates the form of the dataset for lookback = 3.**

```
function SIMULATEDANNEALINGMIN()
    T ← T_max
    best ← INIT()
    while T > T_min do
        next ← NEIGHBOUR(T, best)
        ΔE ← ENERGY(next) − ENERGY(best)
        if ΔE < 0 then
            best ← next
        else if RANDOM() < ACCEPT(T, ΔE) then
            best ← next
        T ← COOLING(T, best)
    return best
```

**Fig 4. Simulated annealing algorithm.**

This likelihood of changing the current solution is also determined by the parameter T (Temperature). Uphill movements are more common with higher T values. Fig 4 shows the simulated annealing algorithm.

Table 1 shows the simulated annealing parameters.

This work extends our previously proposed algorithm, which used a genetic algorithm with LSTM for time step optimization. In that algorithm, we optimized the lookback and the number of neurons in each layer, which took a lot of time. The focus of that work was on improving the performance but not reducing the computation time, so we decided to use simulated annealing to improve the performance (lookback) and reduce the computation time. The success of a genetic algorithm depends on the number of generations and the number of instances in each generation. LSTM has to be executed 100 times for ten generations and ten instances, whereas the simulated annealing requires only a small number of iterations to find an optimal lookback.

**Table 1. This table shows the hyper-parameters for simulated annealing.**

| Simulated annealing parameters | |
| --- | --- |
| Parameters | Values |
| Iterations | 20 |
| Initial temperature | 10 |
| Step Size | 10 or 20 |
| Range/ Bounds | 10-500, 20-500 |

## LSTM

LSTM is a powerful timeseries prediction algorithm used for genetics [23], windpower prediction [24, 25], text processing [26], and human action prediction.

The LSTM comprises three sections, as illustrated in the diagram Fig 5, each of which serves a different function. The first component determines whether the last timestamp's information should be remembered or is irrelevant and can be ignored. The cell attempts to learn new information from the input in the second section. Finally, the cell sends updated information from the current timestamp to the next timestamp in the third component. The gates are the three components of an LSTM cell. The Forget gate is the first component, the Input gate is the second, and the Output gate is the third.

Fig 5 shows the architecture of an LSTM. Eqs 1–3 represents the functions in LSTM cell. F, C, I, and O are the forget, Candidate, Input, and Output gates.

$$f_t = sigmoid(X_t * U_f + H_{t-1} * W_f)$$

$$\bar{C}_t = tanh(X_t * U_c + H_{t-1} * W_c)$$

$$f_t = sigmoid(X_t * U_i + H_{t-1} * W_i)$$

$$f_t = sigmoid(X_o * U_f + H_{t-1} * W_o)$$

$$C_t = f_t * C_{t-1} + I_t * \bar{C}_t$$

$$H_t = O_t * tanh(C_{t-1})$$

$$(1)$$

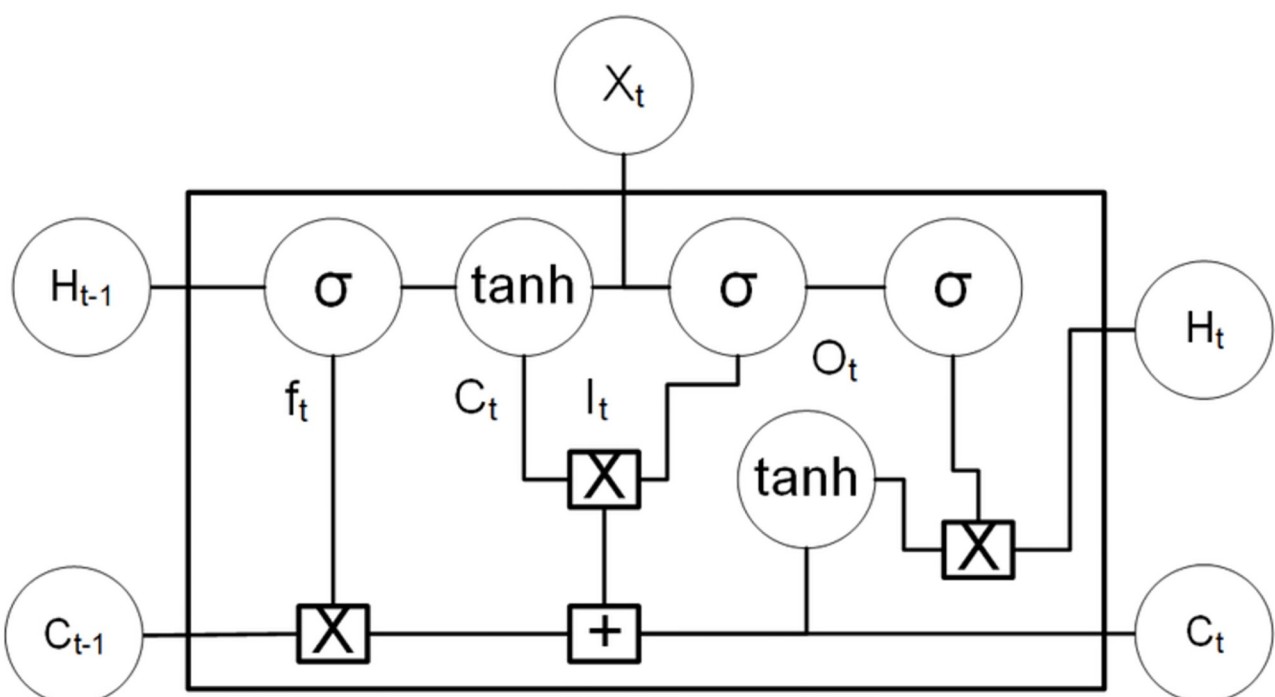

**Fig 5. LSTM architecture.**

$$X_t = \; Input \; Vector$$

$$H_{t-1} = \; Previous \; Cell \; Output$$

$$C_{t-1} = \; Previous \; Cell \; Memory \qquad (2)$$

$$H_t = \; Current \; Cell \; Output$$

$$C_t = \; Current \; Cell \; Memory$$

$$W_f, U_f = \; weight \; vectors \; for \; forget \; gate$$

$$W_c, U_c = \; weight \; vectors \; for \; candidate \; gate$$

$$\qquad (3)$$

$$W_i, U_i = \; weight \; vectors \; for \; input \; gate$$

$$W_o, U_o = \; weight \; vectors \; for \; output \; gate$$

Table 2 shows the LSTM architecture for the deep learning model.

## Simulated annealing objective function

This section discusses and explains the objective function of simulated annealing.

The objective of simulated annealing is to reduce the validation mean square error (See Eq 4), which is calculated using y_validation_data and predicted_values = (LSTM.predict(X_validation_data)).

$$MSE = \left(\frac{1}{n}\right)\sum_{i=1}^{n}(y_i - x_i)^2 \qquad (4)$$

In Eq 4, $n$ represents the number of samples in the validation set, $x$ represents the predicted samples, $y$ represents the actual samples, and $i$ represents the $i^{th}$ instance.

The following text unpacked the objective function, which minimizes the mean square error $MSE$.

1. minimize(MSE)

2. minimize(objectivefunction(lookback))—The objective function takes lookback as an input.

3. minimize(objectivefunction(LSTMtrainingfunction(lookback)))—The objective function takes lookback as an input, trains the LSTM model, and returns the MSE.

**Table 2. This table shows the hyper-parameters for machine learning.**

| Machine learning model architecture | |
| --- | --- |
| Layers | Parameters |
| Layer 1—LSTM | 200 |
| Layer 2—LSTM | 80 |
| Layer 3—LSTM | 50 |
| Layer 4—Dense | 1 |

## The computation cost of the proposed algorithm

The following calculation shows the time for simple LSTM when lookback is increased from 1 to N.

$$\text{Time for lookback} = 1 \quad = O(1)$$
$$\text{Time for lookback} = 2 \quad = O(2)$$
$$\text{Time for lookback} = 3 \quad = O(3)$$
$$\vdots$$
$$\text{Time for lookback} = \text{N} \quad = O(N)$$

Above mentioned equations show the training time for LSTM for lookback 1 to N, and the total time is shown in Eq 5.

$$\sum_1^N (x) = N(N+1)/2 = N^2 \tag{5}$$

The following calculation shows the time when simulated annealing is used to find an optimal lookback. Consider the worst-case scenario in which N is selected for 20 iterations.

$$\text{Time for 20 iterations with lookback} = \text{N} = O(20N) = O(N) \tag{6}$$

Compare Eq 5 with Eq 6. The computation time of Eq 6 (when simulated annealing is used with LSTM for prediction) is far less when lookback is increased from 1 to N. If the number of iterations = N, the cost for both approaches becomes the same.

## Stress analysis on lookback

When simulated annealing is used to find the optimal lookback, we specify a particular range in which simulated annealing should look for the next lookback. Increasing lookback directly affects the dataset's size because this increase results in data replication. Consider the following calculation to understand the size of the dataset in memory as the lookback is increased linearly.

Eq 7 shows the relationship between the size of the dataset and lookback.

$$\text{Total size in memory} = (\text{Total Rows} - \text{lookback}) * (1 + \text{lookback}) \tag{7}$$

- For lookback = 0; total size = $(10 - 0)(1 + 0) = 10$
- For lookback = 1; total size = $(10 - 1)(1 + 1) = 18$
- For lookback = 2; total size = $(10 - 2)(1 + 2) = 24$
- For lookback = 3; total size = $(10 - 3)(1 + 3) = 28$
- For lookback = 4; total size = $(10 - 4)(1 + 4) = 30$
- For lookback = 5; total size = $(10 - 5)(1 + 5) = 30$
- For lookback = 6; total size = $(10 - 6)(1 + 6) = 28$
- For lookback = 7; total size = $(10 - 7)(1 + 7) = 24$
- For lookback = 8; total size = $(10 - 8)(1 + 8) = 18$
- For lookback = 9; total size = $(10 - 9)(1 + 9) = 10$

If we increase the lookback, the size of the dataset increase, so we cannot consider all the lookbacks when using simulated annealing. There must be a specific bound on lookback; otherwise, the system memory would not be able to handle it.

## Results

This section compares each method's computation time and performance: `Naïve LSTM` (A simple LSTM model with lookback = 1 and epochs = 200), `Simple LSTM` (A simple LSTM model with lookback = 1-500 and epochs = 30), and `LSTM with Simulated Annealing` (LSTM model with 20 iterations of simulated annealing, 30 epochs, and lookback = 1-500). We observed that LSTM's performance for windpower prediction could be increased by mutating two parameters: the first one is the number of epochs (the number of times that the learning algorithm will work through the entire training), and the second is the lookback (Number of previous time steps to predict the current time step or the window size in terms of LSTM). For `Naïve LSTM` we considered 500 epochs and loopback = 1 it took 134 seconds yield MSE of about 0.25 for windfarm 1. It was executed quickly, but the difference between the predicted values and actual values was very high, as shown in Table 3.

For `Simple LSTM` we considered epochs = 30 and lookback = [1-500] to find an optimal solution. We considered lookback from 1 to 500 because we do not know where the optimal lookback exists, and to find an optimal lookback iteration, an overall lookback is required. This brute force technique works, but the computation time increases exponentially. The time for epochs = 200 and lookback = 1-500 is 166 hours as shown in Fig 6, and it is extremely computationally expensive. Moreover, the size of the dataset is increased as we increase the lookback, as shown in Fig 7. The difference (MSE, RMSE, r2_score, explained variance, and MAE) between the actual values and predicted values are shown in Fig 8.

The last method is `LSTM with Simulated Annealing` in which we tried to reduce the computation time to find an optimal lookback without affecting the performance. We used simulated annealing in two steps: the first is to find the optimal lookback, and the second is to use the optimal lookback and increase the epochs to 200. For simulated annealing-based LSTM (20 iterations of simulated annealing to find an optimal lookback and 30 epochs), the computation time was 2.67 hours.

In Tables 4–6 columns 2, 3, 4, 5, 6, 7, 8, 9, 10 and 11 show the validation (mse, mae, rmse, explained variance, r2_score) and test (mse, mae, rmse, explained variance, r2_score) respectively for each iteration of simulated annealing. Table 7 shows the final results for 3 wind farms. The MSE for wind farms 1,2 and 3 is reduced to 0.0059,0.0074 and 0.010, respectively.

The two-tailed p-values between each metric when simulated annealing is not used are shown in Table 8.

It is essential to understand overfitting and underfitting when simulated annealing is used for optimization.

**Table 3. This table shows the result for LSTM with epochs = 500 and lookback = 1.**

| Evaluation parameters | Windfarm 1 | Windfarm 2 | Windfarm 3 |
|---|---|---|---|
| MSE: | 0.024947049245487102 | 0.02611486325333724 | 0.02829669487170925 |
| MAE: | 0.11800145355254196 | 0.11815905003994781 | 0.12089438340221412 |
| RMSE: | 0.15794634926292883 | 0.16160093828111655 | 0.16821621465158837 |
| r2_score: | 0.36837154790677196 | 0.5064569510820034 | 0.649444796408535 |
| Explained_variance_score: | 0.3687768353942027 | 0.5064922151886467 | 0.6494470379350865 |

The first, second, and third columns show the results for windfarm 1,2, and 3, respectively.

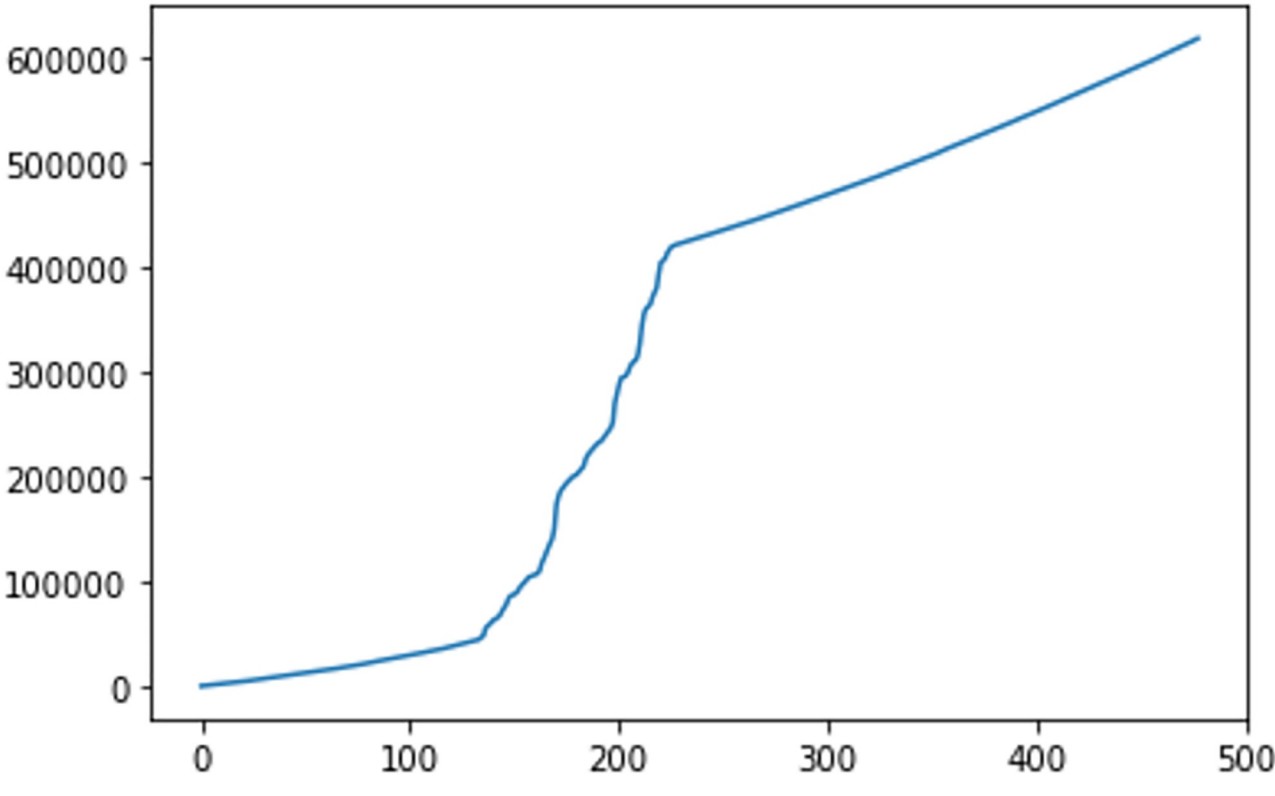

**Fig 6. This figure shows the computation time in seconds for lookback 1 to 500 and 30 epochs.**

The overfitting and underfitting of simulated annealing can be inferred from the results. In the case of simulated annealing, the optimal lookback can be a sub-optimal solution, but whether the optimal/sub-optimal lookback overfits/underfits on the training or not can be inferred from the test MSE.

The test data is not used to find an optimal look back, so the LSTM performance without simulated annealing and with simulated annealing can be used to see whether the model overfits or not on the training data. So, we see that MSE on test data is reduced with simulated annealing, which means the model does not overfit the training data when simulated annealing is used to find an optimal lookback. As far underfitting is concerned, the model does not underfit because when optimal lookback is used to train the model, the training MSE is reduced compared to when simulated annealing is not used.

## Discussion

This section discusses the results generated by all two algorithms: `Simple LSTM` (A simple LSTM model with lookback = 1-500 and epochs = 30), and `LSTM with Simulated Annealing` (LSTM model with 20 iterations of simulated annealing, 30 epochs, and lookback = 1-500).

The first interesting observation is reduced computation time. Consider the results for windfarm 1 Table 4. The simulated annealing algorithm starts with lookback = 220, increasing the iterations. It mutates the lookback, rearranges the training data accordingly, and reports the evaluation metrics. The results for lookback = 236, shown in row 8, are the best, so we used that lookback to train the model for 500 epochs. Compare the same process with the simple

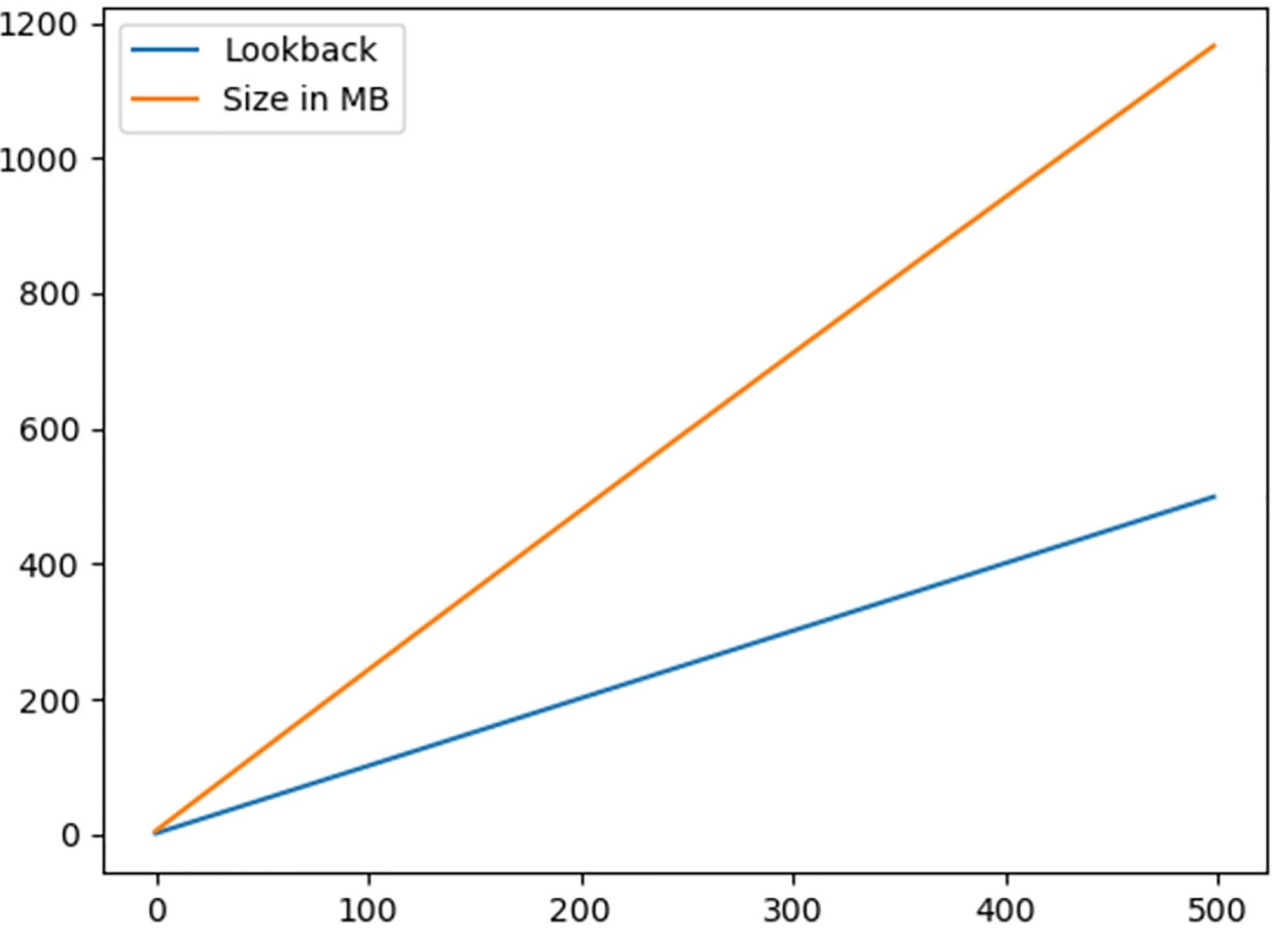

**Fig 7. This diagram shows the relationship between data size and lookback.** As we increase the lookback, the size of the data increases exponentially.

LSTM, which iterates from lookback 1 to 500 and reports the epochs as shown in the Fig 8. The computation is the cost high in the simple LSTM.

Second, lookback for wind farms 1, 2, and 3 are different because the lookback is optimized for a particular wind farm. Third, we considered minimizing mean absolute error as an objective function. However, minimization of other parameters like mean explained variance or mean square error can also be considered an objective function.

The computation cost of the whole process depends on the number of iterations of the simulated annealing. Consider the number of epochs = 30, lookback in the range [1-500], and 20 iterations of simulated annealing. For each iteration, the computational time will be different because it depends on the lookback, as shown in the diagram Fig 6. The max time for each iteration is within the range [Time algorithm takes for lookback 1 = $T1$—Time algorithm takes for lookback 500 = $T500$]. The maximum time for 20 iterations can be $20^*T500$ if simulated annealing chooses lookback = 500 for all 20 iterations.

The time algorithm takes when lookback is linearly increased from 1 to 500 is the sum of all look back, which is greater than $20^*T500$. After finding the best lookback, the final step is to use LSTM with the best lookback for a specific number of epochs. The number of epochs in the final iteration is greater than the number of epochs used for the simulated annealing. So that is how simulated annealing reduces the execution time without reducing the performance.

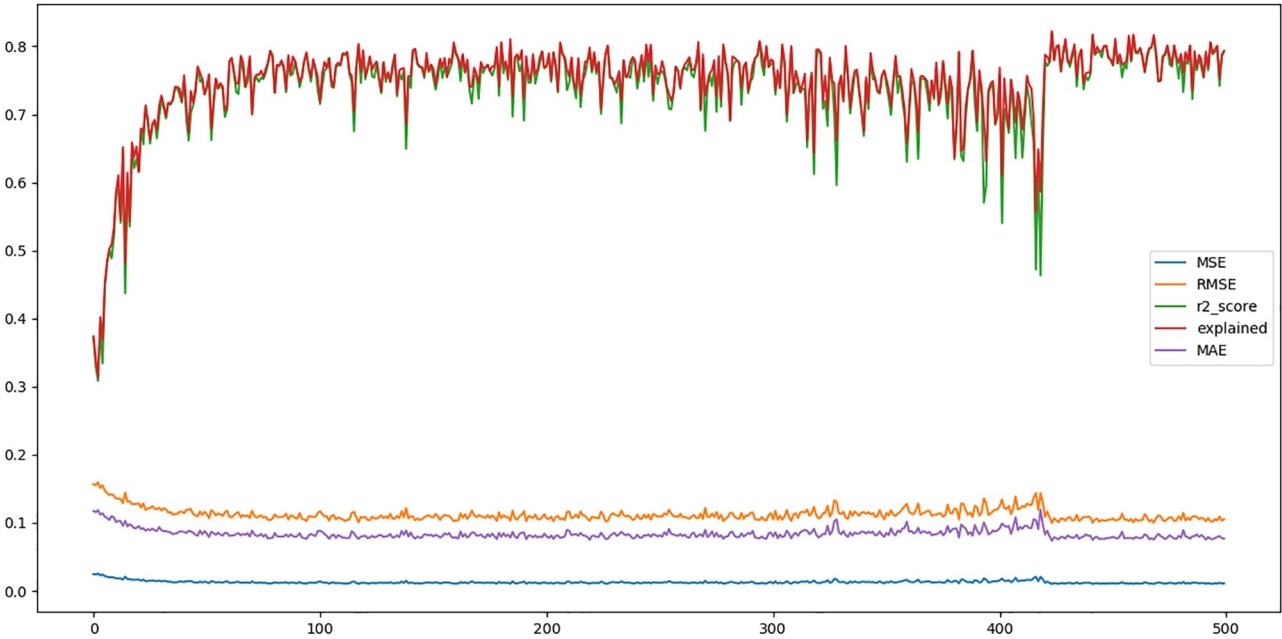

**Fig 8. This figure shows the results for 200 epochs at each lookback starting from 1 to 500.** MSE decreases as we increase the lookback.

## Conclusion

This section includes the concluding remarks, future directions, and the limitation of the proposed algorithm. In this article, we proposed a simulated and healing-based LSTM for wind power prediction, which reduces the time to find an optimal lookback for LSTM prediction reducing the loss function. Rather than using simulated annealing, a particle swarm optimization algorithm or a combination of genetic algorithms and simulated annealing can be used for more robust predictions.

Due to limited space, we considered only 500 lookbacks (2009-07-02-13 to 2009-07-23-07 = 21 previous days), but in reality, data for more than one month should be considered to make the current prediction, which we believe can significantly increase the prediction. It also depends on the season; for example, the wind velocity is not constant throughout a year for a particular area, so if we consider large lookback information, the model's performance may degrade.

We considered wind power prediction, and in the future, we plan to do it for other forecasting problems related to the energy sector like solar power prediction. Second, we considered only one optimization algorithm, simulated annealing, but optimization algorithms like particle swarm optimization and genetic algorithm can also be used to benchmark the performance. Benchmarking is necessary because different optimization algorithms yield different optimal lookbacks, and simulated annealing tries only a limited number of lookbacks (same as the number of iterations) to find an optimal one. Third, we considered only the lookback parameter for optimization, but simulated annealing can also be used to optimize multiple parameters like the number of epochs, which will undoubtedly increase the overall computation cost. The LSTM can be replaced with BILSTM, GRU, and machine learning-based regression algorithms with minor modifications.

Following are the computer and library specs used for implementing models and generating results. The system specifications are: Intel(R) Core(TM) 7-9750H CPU @ 2.60Hz, 16 GB

**Table 4. Iteration 8 for lookback = 236 yields the minimum validation evaluation metrics.**

| Iteration | vmse | vmae | vrmse | vexplained variance Score | vr2_Score | tmse | tmae | trmse | texplained variance Score | tr2_Score | Lookback |
|---|---|---|---|---|---|---|---|---|---|---|---|
| 0 | 0.013555286 | 0.089657726 | 0.116427169 | 0.718714019 | 0.722873228 | 0.014197587 | 0.089082477 | 0.119153626 | 0.718098144 | 0.722106001 | 220 |
| 1 | 0.013425251 | 0.087130105 | 0.115867387 | 0.734633907 | 0.735483667 | 0.013393207 | 0.087018675 | 0.115729025 | 0.725576734 | 0.726122176 | 231 |
| 2 | 0.013309718 | 0.086404348 | 0.11536775 | 0.713879529 | 0.716479277 | 0.013249369 | 0.087034575 | 0.115105903 | 0.69219984 | 0.694500305 | 226 |
| 3 | 0.014955238 | 0.092924894 | 0.122291612 | 0.705921906 | 0.713572044 | 0.015235962 | 0.093165822 | 0.123434039 | 0.686088975 | 0.696947576 | 232 |
| 4 | 0.014100669 | 0.088821803 | 0.118746238 | 0.753824502 | 0.756825462 | 0.014496925 | 0.089811982 | 0.120403178 | 0.737770036 | 0.741399416 | 224 |
| 5 | 0.01339108 | 0.08587575 | 0.115719833 | 0.704181792 | 0.704589972 | 0.012644675 | 0.083997946 | 0.112448546 | 0.733487877 | 0.735011383 | 230 |
| 6 | 0.015167523 | 0.090565598 | 0.123156498 | 0.650412247 | 0.668460655 | 0.014874994 | 0.088230279 | 0.121963085 | 0.649017814 | 0.665477287 | 233 |
| 7 | **0.011620712** | **0.081698937** | **0.107799409** | **0.748759165** | **0.751719186** | **0.012320427** | **0.083440777** | **0.110997419** | **0.733035736** | **0.734815168** | **236** |
| 8 | 0.014044705 | 0.090035525 | 0.118510359 | 0.699363023 | 0.699425789 | 0.012726525 | 0.085709828 | 0.112811903 | 0.720914788 | 0.721259475 | 236 |
| 9 | 0.014080487 | 0.090859153 | 0.118661226 | 0.688758934 | 0.698402331 | 0.013298642 | 0.089160635 | 0.115319737 | 0.702843344 | 0.715129513 | 249 |
| 10 | 0.012528327 | 0.082828924 | 0.111930009 | 0.742812724 | 0.744651183 | 0.011984276 | 0.080976985 | 0.10947272 | 0.765855216 | 0.76714211 | 260 |
| 11 | 0.013443297 | 0.085318941 | 0.115945233 | 0.728286742 | 0.732141311 | 0.014040075 | 0.087120903 | 0.118490822 | 0.708010447 | 0.712024182 | 250 |
| 12 | 0.013920278 | 0.09223864 | 0.117984228 | 0.705207655 | 0.725969821 | 0.013578309 | 0.091058571 | 0.116526001 | 0.718249835 | 0.744989242 | 249 |
| 13 | 0.012608901 | 0.082293553 | 0.112289362 | 0.741317054 | 0.749595772 | 0.012297802 | 0.080862145 | 0.110895455 | 0.757738097 | 0.762325163 | 246 |
| 14 | 0.013363067 | 0.084547985 | 0.115598731 | 0.706377367 | 0.719273487 | 0.01377071 | 0.086268398 | 0.117348669 | 0.698726696 | 0.713113069 | 235 |
| 15 | 0.013308869 | 0.087768001 | 0.115364072 | 0.675429322 | 0.676449757 | 0.01270307 | 0.086174591 | 0.112707896 | 0.693419655 | 0.694314747 | 228 |
| 16 | 0.013827367 | 0.090194503 | 0.117589825 | 0.735399509 | 0.74422925 | 0.01515852 | 0.092890986 | 0.123119941 | 0.70444375 | 0.713699532 | 223 |
| 17 | 0.012293933 | 0.08294426 | 0.110878008 | 0.751845609 | 0.753574427 | 0.012683574 | 0.084163177 | 0.112621374 | 0.74437856 | 0.746130732 | 212 |
| 18 | 0.012023005 | 0.081782421 | 0.109649464 | 0.760631615 | 0.761180689 | 0.012668695 | 0.083787024 | 0.112555296 | 0.741519925 | 0.742407603 | 203 |
| 19 | 0.01280214 | 0.084426416 | 0.113146544 | 0.719464222 | 0.730930099 | 0.012670585 | 0.081849585 | 0.112563691 | 0.728481838 | 0.736308141 | 193 |
| 20 | 0.012514733 | 0.084945384 | 0.111869268 | 0.767739182 | 0.782057068 | 0.013139883 | 0.087568296 | 0.11462933 | 0.732854861 | 0.748849039 | 172 |

**Table 5. Iteration 0 for lookback = 221 yields the minimum validation evaluation metrics.**

| Iteration | vmse | vmae | vrmse | vexplained variance Score | vr2_Score | tmse | tmae | trmse | texplained variance Score | tr2_Score | Lookback |
|---|---|---|---|---|---|---|---|---|---|---|---|
| 0 | **0.014021133** | **0.084809543** | **0.118410864** | **0.79040578** | **0.793697823** | **0.015356417** | **0.090398555** | **0.123921009** | **0.751978074** | **0.756690967** | **221** |
| 1 | 0.015981298 | 0.090843498 | 0.12641716 | 0.710750064 | 0.734523997 | 0.015761335 | 0.089975634 | 0.125544154 | 0.726164072 | 0.747601373 | 231 |
| 2 | 0.018307923 | 0.096850453 | 0.135306773 | 0.686247451 | 0.711666477 | 0.016530055 | 0.09357702 | 0.128569263 | 0.710866251 | 0.728784288 | 226 |
| 3 | 0.016666265 | 0.095417525 | 0.129097891 | 0.717642869 | 0.717880153 | 0.015692573 | 0.091560269 | 0.125270001 | 0.737398688 | 0.73760531 | 230 |
| 4 | 0.016528089 | 0.091911691 | 0.128561615 | 0.721457283 | 0.741416073 | 0.016735208 | 0.092720059 | 0.129364632 | 0.717395221 | 0.739502181 | 229 |
| 5 | 0.015416824 | 0.090627607 | 0.124164503 | 0.742319884 | 0.742583982 | 0.015701013 | 0.09160624 | 0.125303684 | 0.755269612 | 0.755298188 | 226 |
| 6 | 0.016941045 | 0.09447416 | 0.130157769 | 0.722902687 | 0.722908668 | 0.016499551 | 0.092423636 | 0.128450579 | 0.722987457 | 0.723155077 | 233 |
| 7 | 0.015243358 | 0.092560218 | 0.123463995 | 0.70276444 | 0.703090637 | 0.015535459 | 0.092143498 | 0.124641321 | 0.699223051 | 0.699225531 | 236 |
| 8 | 0.016400691 | 0.092566441 | 0.128065182 | 0.72524384 | 0.733550152 | 0.015019684 | 0.089691451 | 0.122554821 | 0.754714312 | 0.760084318 | 236 |
| 9 | 0.01673071 | 0.094486272 | 0.129347248 | 0.742337437 | 0.746699027 | 0.016370678 | 0.094179436 | 0.12794795 | 0.741700362 | 0.74769193 | 249 |
| 10 | 0.019308244 | 0.104275267 | 0.138954106 | 0.695842805 | 0.710310531 | 0.018774894 | 0.102183051 | 0.137021511 | 0.70034405 | 0.713266965 | 260 |
| 11 | 0.014157685 | 0.085643589 | 0.11898607 | 0.771687765 | 0.772193331 | 0.013822716 | 0.086157021 | 0.117570046 | 0.775519517 | 0.775680332 | 256 |
| 12 | 0.016018131 | 0.091373796 | 0.126562754 | 0.755624857 | 0.756174114 | 0.016203707 | 0.091968548 | 0.127293783 | 0.747380788 | 0.747684814 | 277 |
| 13 | 0.016732945 | 0.096212232 | 0.129355885 | 0.746778934 | 0.768427322 | 0.016548652 | 0.096444054 | 0.128641565 | 0.752005955 | 0.77248675 | 276 |
| 14 | 0.018185315 | 0.09703289 | 0.134852937 | 0.658156786 | 0.681191311 | 0.01671561 | 0.09385008 | 0.129288862 | 0.685320212 | 0.706647075 | 279 |
| 15 | 0.017939203 | 0.097697472 | 0.133937309 | 0.682664011 | 0.69953171 | 0.016481592 | 0.092903353 | 0.128380654 | 0.712629777 | 0.727334455 | 279 |
| 16 | 0.014802583 | 0.090809952 | 0.121665867 | 0.766994253 | 0.7700836 | 0.015928004 | 0.093258772 | 0.126206197 | 0.753956017 | 0.755718438 | 283 |
| 17 | 0.015422492 | 0.090421909 | 0.124187326 | 0.735615615 | 0.742374076 | 0.016291272 | 0.091742128 | 0.127637267 | 0.740540697 | 0.748009552 | 289 |
| 18 | 0.020577146 | 0.109819262 | 0.143447365 | 0.662520475 | 0.703781516 | 0.020691314 | 0.110542513 | 0.143844758 | 0.658580929 | 0.696268672 | 291 |
| 19 | 0.014668267 | 0.090077138 | 0.121112621 | 0.743525939 | 0.745375497 | 0.015314464 | 0.091970547 | 0.123751622 | 0.738708752 | 0.739562733 | 284 |
| 20 | 0.015187367 | 0.092814131 | 0.123237037 | 0.77659827 | 0.787984041 | 0.016308332 | 0.094902923 | 0.127704078 | 0.760574737 | 0.77445938 | 274 |

**Table 6. Iteration 1 for lookback = 231 yields the minimum validation evaluation metrics.**

| Iteration | vmse | vmae | vrmse | vexplained variance Score | vr2_Score | tmse | tmae | trmse | texplained variance Score | tr2_Score | Lookback |
|---|---|---|---|---|---|---|---|---|---|---|---|
| 0 | 0.018139678 | 0.096378329 | 0.134683622 | 0.784061983 | 0.791991646 | 0.018631792 | 0.097262435 | 0.136498322 | 0.780126281 | 0.787187 | 221 |
| 1 | **0.016843123** | **0.094559559** | **0.129781058** | **0.794449977** | **0.794709819** | **0.016082535** | **0.092499203** | **0.126816934** | **0.800342931** | **0.800609891** | **231** |
| 2 | 0.018535373 | 0.102264359 | 0.136144675 | 0.795880506 | 0.801548697 | 0.018709524 | 0.101428653 | 0.136782761 | 0.780482758 | 0.788560782 | 226 |
| 3 | 0.019264264 | 0.101270419 | 0.138795764 | 0.759460017 | 0.759726252 | 0.018479574 | 0.099190272 | 0.135939596 | 0.771934678 | 0.771949656 | 234 |
| 4 | 0.021230551 | 0.109796556 | 0.145707074 | 0.778464655 | 0.796520016 | 0.021073828 | 0.108475019 | 0.145168274 | 0.774558816 | 0.794103866 | 242 |
| 5 | 0.018148757 | 0.098484646 | 0.134717322 | 0.796176651 | 0.800645958 | 0.018606933 | 0.100127355 | 0.136407231 | 0.788057825 | 0.792167542 | 245 |
| 6 | 0.019408559 | 0.103795839 | 0.139314604 | 0.754895357 | 0.75507525 | 0.019932497 | 0.10422733 | 0.141182496 | 0.743777595 | 0.743786511 | 253 |
| 7 | 0.021746576 | 0.104869068 | 0.147467202 | 0.747498667 | 0.771411081 | 0.021072605 | 0.102670283 | 0.145164061 | 0.754459116 | 0.783665906 | 258 |
| 8 | 0.018420542 | 0.099029868 | 0.135722298 | 0.76802668 | 0.77025322 | 0.018982184 | 0.099260277 | 0.137775847 | 0.763175823 | 0.764571515 | 258 |
| 9 | 0.020698519 | 0.108471742 | 0.143869798 | 0.775522754 | 0.795570938 | 0.020347255 | 0.107094994 | 0.142643805 | 0.778569652 | 0.801528447 | 262 |
| 10 | 0.019716548 | 0.10325018 | 0.140415625 | 0.749780462 | 0.750164842 | 0.019597491 | 0.103638156 | 0.13999104 | 0.744789223 | 0.744847578 | 269 |
| 11 | 0.019069671 | 0.100298775 | 0.138092978 | 0.76690867 | 0.77935028 | 0.019133096 | 0.100519898 | 0.138322433 | 0.770123402 | 0.783598673 | 264 |
| 12 | 0.018170074 | 0.097955461 | 0.134796418 | 0.790902888 | 0.790987336 | 0.018942528 | 0.099051277 | 0.137631856 | 0.780053578 | 0.780058562 | 276 |
| 13 | 0.019161085 | 0.099396417 | 0.138423573 | 0.765950462 | 0.775468262 | 0.01948011 | 0.100511505 | 0.139571164 | 0.758383093 | 0.77326203 | 284 |
| 14 | 0.019987972 | 0.107552859 | 0.141378823 | 0.774375403 | 0.797168826 | 0.020170419 | 0.109261606 | 0.142022599 | 0.755580667 | 0.782221186 | 284 |
| 15 | 0.018507612 | 0.101990537 | 0.136042683 | 0.777753378 | 0.785226889 | 0.017751395 | 0.099731846 | 0.133234361 | 0.788911652 | 0.795042823 | 277 |
| 16 | 0.019420294 | 0.101413537 | 0.139356714 | 0.752847414 | 0.765093819 | 0.019566397 | 0.10078472 | 0.139879938 | 0.747163183 | 0.761579724 | 275 |
| 17 | 0.019988727 | 0.109301142 | 0.141381493 | 0.759387788 | 0.794452815 | 0.021466122 | 0.113304777 | 0.146513214 | 0.750918964 | 0.786228101 | 273 |
| 18 | 0.016938818 | 0.093347283 | 0.130149214 | 0.811011419 | 0.811123305 | 0.018544361 | 0.097225424 | 0.136177683 | 0.787178726 | 0.787737707 | 282 |
| 19 | 0.018730242 | 0.101545649 | 0.136858474 | 0.77716989 | 0.77834573 | 0.017985014 | 0.099443442 | 0.134108219 | 0.77798847 | 0.780613129 | 279 |
| 20 | 0.019617002 | 0.103688755 | 0.140060709 | 0.776362899 | 0.786121423 | 0.020271706 | 0.104626294 | 0.142378743 | 0.764498858 | 0.771318499 | 283 |

**Table 7. This table shows the final result for 200 epochs and the optimal time step found using simulated annealing.**

| Evaluation parameters | Windfarm 1 | Windfarm 2 | Windfarm 3 |
|---|---|---|---|
| MSE: | 0.00598448861289919 | 0.0074024574253095015 | 0.010542748599023146 |
| MAE: | 0.05539247013060056 | 0.06043025969591132 | 0.07200711010548518 |
| RMSE: | 0.07735947655523007 | 0.08603753497927229 | 0.10267788758551252 |
| r2_score: | 0.8938389214978747 | 0.8988519777606736 | 0.8872896132370566 |
| Explained_variance_score: | 0.8943655779202052 | 0.8988558011724342 | 0.8873497151846284 |

The first, second, and third columns show the results for windfarm 1,2, and 3, respectively.

**Table 8. This table shows the two-tailed p-values.**

| Evaluation Metrics | The two-tailed P value |
|---|---|
| MSE | 0.0004 |
| MAE | 0.0004 |
| RMSE | 0.0008 |
| r2_score | 0.0090 |
| Explained_variance_score | 0.0090 |

This shows the two-tailed p-value between both results (without optimization algorithm and with simulated annealing) for all metrics and each wind farm. Values from Tables 3 and 7 were used to find the two-tailed p-values.

RAM as well as a NVIDIA GeForce RTX 2060 GPU, running Microsoft Windows 10. Moreover, the development specifications are: Cuda compilation tools release 10.0, V10.0.130, Deep Learning framework Keras 2.4.3, Python 3.6.8, and Tensorflow 2.3.1.

All the code and dataset used to demonstrate the metholodgy of this paper is available at this link. `NaiveLSTM` directory contains a simple LSTM code (window size 1 and 500 epochs) for wind power prediction for 3 datasets. `SimpleLSTM` directory contains a simple LSTM code (lookback [1-500] and 200 epochs) for wind power prediction for 3 datasets. `SimulatedAnnealing/code—find optimal lookback.py` file contains a simulated annealing LSTM code (to find optimal lookback and 30 epochs) for wind power prediction for 3 datasets. `SimulatedAnnealing/code—use optimal lookback.py` file contains a simulated annealing LSTM code (to find optimal lookback and 30 epochs) for wind power prediction for 3 datasets.

## Author Contributions

**Conceptualization:** Muhammad Muneeb.

**Methodology:** Muhammad Muneeb.

**Software:** Muhammad Muneeb.

**Validation:** Muhammad Muneeb.

**Visualization:** Muhammad Muneeb.

**Writing – original draft:** Muhammad Muneeb.

**Writing – review & editing:** Muhammad Muneeb.

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
