## [Decision Letter · Decision Letter 0]

10 Jul 2022

PONE-D-22-05787LSTM input timestep optimization using simulated annealing for wind power predictionPLOS ONE

Dear Dr. Muneeb,

Thank you for submitting your manuscript to PLOS ONE. After careful consideration, we feel that it has merit but does not fully meet PLOS ONE’s publication criteria as it currently stands. Therefore, we invite you to submit a revised version of the manuscript that addresses the points raised during the review process.

We look forward to receiving your revised manuscript.

Kind regards,

Seyedali Mirjalili

Academic Editor

PLOS ONE

“Unfunded studies”

d) If you did not receive any funding for this study, please state: “The authors received no specific funding for this work.

5. "Acknowledgments Section: Move New Information to the Financial Disclosure:

""Thank you for stating the following in the Acknowledgments Section of your manuscript:

“This publication is based upon work supported by the Khalifa University of

Science and Technology under Award No. CIRA-2019-050 to SFF.”

“Unfunded studies”

6. Thank you for stating the following in your Competing Interests section: 

“NO authors have competing interests”

Reviewers' comments:

Reviewer's Responses to Questions

**Comments to the Author**

1. Is the manuscript technically sound, and do the data support the conclusions?

Reviewer #1: Partly

Reviewer #2: Yes

2. Has the statistical analysis been performed appropriately and rigorously? 

Reviewer #1: N/A

Reviewer #2: Yes

3. Have the authors made all data underlying the findings in their manuscript fully available?

Reviewer #1: Yes

Reviewer #2: Yes

4. Is the manuscript presented in an intelligible fashion and written in standard English?

Reviewer #1: Yes

Reviewer #2: Yes

5. Review Comments to the Author

Reviewer #1: After carefully readying and analysis of the paper, my comments as as follows:

1. The numerical method used in this paper should be investigated further using more statistical techniques.

2. I didn't find in-depth discussion around the computational cost of the proposed method.

3. The results should be discussed further at the end of the results sections to show the reasoning of the authors.

4. In terms of case-study, I wonder if authors can provide justifications on the suitability to stress test the proposed method.

5. I also would like the authors to discuss about two things around the optimisation method, overfitting and underfitting.

6. I also would like the authors to discuss about two things around the optimisation method, bias and model drift.

7. The results should be further explained to demonstrate that accurately proof/disproof the hypothesis.

8. Please highlight the hypothesis of this work in the text.

Reviewer #2: The manuscript, in its present form, contains several weaknesses. Adequate revisions to the following points should be undertaken to justify the recommendation for publication.

1: please add future work to the conclusion section and discuss it briefly.

2. The contribution is not stated also add at the end of the introduction section.

3. What is the main reason for choosing the simulated annealing algorithm?

4. The results and discussion section has to be improved, where more details of the achieved results should be stated clearly in this section. In addition, authors also have to provide some insight discussion of the results.

5. Please explain the objective function mathematically in a separate section.

6. PLOS authors have the option to publish the peer review history of their article (what does this mean?). If published, this will include your full peer review and any attached files.

Reviewer #1: No

Reviewer #2: No

---

## [Decision Letter · Decision Letter 1]

21 Sep 2022

LSTM input timestep optimization using simulated annealing for wind power prediction

PONE-D-22-05787R1

Dear Dr. Muneeb,

We’re pleased to inform you that your manuscript has been judged scientifically suitable for publication and will be formally accepted for publication once it meets all outstanding technical requirements.

Kind regards,

Seyedali Mirjalili

Academic Editor

PLOS ONE

Additional Editor Comments (optional):

Reviewers' comments:

Reviewer's Responses to Questions

**Comments to the Author**

1. If the authors have adequately addressed your comments raised in a previous round of review and you feel that this manuscript is now acceptable for publication, you may indicate that here to bypass the “Comments to the Author” section, enter your conflict of interest statement in the “Confidential to Editor” section, and submit your "Accept" recommendation.

Reviewer #2: All comments have been addressed

2. Is the manuscript technically sound, and do the data support the conclusions?

Reviewer #2: Yes

3. Has the statistical analysis been performed appropriately and rigorously? 

Reviewer #2: Yes

4. Have the authors made all data underlying the findings in their manuscript fully available?

Reviewer #2: Yes

5. Is the manuscript presented in an intelligible fashion and written in standard English?

Reviewer #2: Yes

6. Review Comments to the Author

Reviewer #2: The authors did great efforts in response to the comments given in the previous round of review. The paper is acceptable now.

7. PLOS authors have the option to publish the peer review history of their article (what does this mean?). If published, this will include your full peer review and any attached files.

Reviewer #2: No

---

## [Editor Report · Acceptance letter]

29 Sep 2022

PONE-D-22-05787R1 

LSTM input timestep optimization using simulated annealing for wind power predictions 

Dear Dr. Muneeb:

I'm pleased to inform you that your manuscript has been deemed suitable for publication in PLOS ONE. Congratulations! Your manuscript is now with our production department. 

Kind regards, 

on behalf of

Prof. Seyedali Mirjalili 

Academic Editor

PLOS ONE